# Oxylipins and Reactive Carbonyls as Regulators of the Plant Redox and Reactive Oxygen Species Network under Stress

**DOI:** 10.3390/antiox12040814

**Published:** 2023-03-27

**Authors:** Madita Knieper, Andrea Viehhauser, Karl-Josef Dietz

**Affiliations:** Biochemistry and Physiology of Plants, Faculty of Biology, Bielefeld University, 33615 Bielefeld, Germany

**Keywords:** 12-oxophytodienoic acid, oxylipin, redox regulation, reactive oxygen species

## Abstract

Reactive oxygen species (ROS), and in particular H_2_O_2_, serve as essential second messengers at low concentrations. However, excessive ROS accumulation leads to severe and irreversible cell damage. Hence, control of ROS levels is needed, especially under non-optimal growth conditions caused by abiotic or biotic stresses, which at least initially stimulate ROS synthesis. A complex network of thiol-sensitive proteins is instrumental in realizing tight ROS control; this is called the redox regulatory network. It consists of sensors, input elements, transmitters, and targets. Recent evidence revealed that the interplay of the redox network and oxylipins–molecules derived from oxygenation of polyunsaturated fatty acids, especially under high ROS levels–plays a decisive role in coupling ROS generation and subsequent stress defense signaling pathways in plants. This review aims to provide a broad overview of the current knowledge on the interaction of distinct oxylipins generated enzymatically (12-OPDA, 4-HNE, phytoprostanes) or non-enzymatically (MDA, acrolein) and components of the redox network. Further, recent findings on the contribution of oxylipins to environmental acclimatization will be discussed using flooding, herbivory, and establishment of thermotolerance as prime examples of relevant biotic and abiotic stresses.

## 1. Introduction

About 2.4 billion years ago, molecular oxygen was first introduced into the earth’s atmosphere by oxygenic photosynthesis [1]. Alongside respiratory and photosynthetic electron transport and relevant enzyme activities, reactive oxygen species (ROS), derivatives of O_2_, originated as permanently formed byproducts of metabolism. To survive and grow under fluctuating environmental conditions, especially in the context of ongoing climate change and global warming, plants have evolved a complex system of molecular mechanisms and pathways of sensing, transmitting, and responding to optimal acclimatization [2,3]. The key components are ROS, in particular superoxide anions (O_2_^•−^) and hydroxyl radicals (^•^OH), which are free radicals containing an unpaired electron of varying reactivity, singlet oxygen (^1^O_2_), which is an excited non-radical derived from molecular oxygen by spin inversion, and hydrogen peroxide (H_2_O_2_), originating from O_2_^•−^ dismutation or directly from two electron transfer reactions. These components accumulate under suboptimal growth conditions [2,4,5,6]. Generation of ROS occurs in plastids, peroxisomes, endoplasmic reticulum (ER), mitochondria, apoplast, and cytosol [2,3,4,5,6,7]. While they are generated constantly by housekeeping enzymes and as byproducts of metabolism, biotic and abiotic stresses enhance ROS synthesis distinctly [2,4,8]. Overall, 1–2% of the total oxygen consumed by plants has been proposed to account for ROS formation [9,10,11,12].

On the one hand, ROS serve as second messengers, modulating stress defense, hormone signaling, development, and growth processes. Resting H_2_O_2_ concentrations in plasmatic compartments are in the range of 10–30 nM [13,14]. On the other hand, ROS are highly reactive, hence posing a significant oxidative threat at high concentrations [2,4,5,6,7,15]. Therefore, plant cells monitor and modulate ROS levels vigorously to maintain their signaling function while preventing oxidative damage. The basic processes of this modulation of both ROS and all reactive molecular species (RMS) include regulation of (1) RMS synthesis, (2) RMS sensing and signal processing, and (3) RMS degradation [2,16,17]. These processes are implemented by the redox-regulatory network, which controls the total antioxidant capacity of the cell.

## 2. The Plant ROS and Redox Network

ROS homeostasis as influenced by ROS generation on the one hand and decomposition on the other shapes the way ROS affect plant cells and will therefore constitute the first chapter of this review. Subsequently, oxylipins, products of ROS-induced disruption of lipid membranes, will be introduced and their influence on the redox network characterized (Figure 1).

### 2.1. Synthesis of ROS in Higher Plants

The major organelles contributing to stress-induced ROS formation are chloroplasts, followed by mitochondria due to their electron transport chains [2,6,11]. For instance, the electron transport chain of cellular respiration is leaky, with an estimated percentage of 0.1% to 2% of electrons passing the chain being released, thereby causing generation of O_2_^•−^ [5]. Plastid generation of O_2_**^•^**^−^ is achieved by the univalent reduction of molecular O_2_, primarily in the photosystem I [2,6]. O_2_^•−^ does not only serve as a precursor for other ROS; it also impedes ROS scavenging by antioxidant enzymes (e.g., peroxidases) [6,8,11,18]. Further, regarding plastid ROS synthesis, ^1^O_2_, another common and highly reactive ROS, is produced in both photosystems by transfer of energy from excited chromophores to O_2_ [6]. In the apoplast, O_2_^•-^ and H_2_O_2_ are formed by respiratory burst oxidase homologues (RBOHs), polyamine oxidases, pH-dependent peroxidases, and copper amine oxidases. O_2_^•−^, which is unstable and non-permeable to membranes, further undergoes protonation and dismutation and hence contributes to the extracellular H_2_O_2_ pool [5]. H_2_O_2_ is imported into the cell by aquaporins [2]. Additionally, H_2_O_2_ is directly produced by SOD-mediated or spontaneous dismutation of O_2_^•−^ and by enzyme-catalyzed two electron transfer, e.g., by xanthine or glycolate oxidases, the latter as part of peroxisomal photorespiration [2,4,6]. As photorespiration is a major source of H_2_O_2_, contributing about 70% of generated H_2_O_2_ (as studied in wheat with a supposed flux of 10% of photosynthetic electrons to photorespiration), peroxisomes contribute distinctly to cellular oxidative metabolism [19,20,21]. Besides H_2_O_2_, peroxisomal metabolism supplies essential signaling molecules, such as reactive nitrogen species including nitric oxide (NO) and jasmonic acid (JA, see Section 3.1.2) [20]. More detailed insights into peroxisomal metabolism can be found in other reviews [20,21,22,23]. 

ROS can act locally, but they can also be transported to different organelles and serve as signals with maximal migration distances ranging from 1 nm (^•^OH) to more than 1 µm (H_2_O_2_) [8,16]. Their levels in different cell compartments is adjusted by local synthesis in combination with import, export, and degradation differ between different stress conditions, creating unique ROS signatures [4].

### 2.2. ROS Homeostasis as Key Mechanism to Avoid Oxidative Stress

As oxidizing agents, ROS react with proteins, especially reduced and deprotonated thiolate residues. This alters catalytic activity of thiol-switch proteins which are essential in a plethora of plant processes, including metabolism and regulation of the antioxidant pool. In the case of H_2_O_2_, one of the most reactive ROS (only ^•^OH, the hydroxyl radical, is more reactive), reaction with thiolates leads to generation of sulfenic, sulfinic, and sulfonic acid derivatives; only the first two of these can be reversed (non-)enzymatically [2,8]. Furthermore, ROS can fragmentize peptide chains and increase protein aggregation and degradation [5]. As oxidation of proteins by ROS is essential for ROS sensing and signaling, a highly complex network of oxidation-sensitive proteins, the thiol-redox network, is dedicated to modulating ROS function and level. This network will further be characterized in Section 2.2.1.

Additional targets of ROS-induced oxidative damage are DNA and RNA; these are prone to modification of nucleotide bases and breakage of single- and double-strands, e.g., by rupture of nucleosomes [8,11].

Finally, they are linked to lipid peroxidation, thereby contributing to synthesis of oxylipins. Oxylipins are regulatory compounds derived from the oxidation of polyunsaturated fatty acids (PUFA) [8]. As ROS synthesis often occurs close to membranes, lipids are regarded as the primary target of ROS [9,24]. More detailed information on lipid peroxidation, subsequent formation of oxylipins, and their highly diverse functions in basic plant processes will be provided in Section 3 and 4 of this review.

#### 2.2.1. Structure of the Redox-Regulatory Network

In general, the redox network can be divided hierarchically into several regulatory levels, starting with electron flow from metabolic and photosynthetic activity to redox input elements such as ferredoxin, NADPH, and γ-glutamyl-cysteinyl-glycine (glutathione, GSH). These input elements transfer electrons to redox transmitters, involving dedicated enzymes such as ferredoxin-thioredoxin reductases [25]. Redox transmitters reduce ROS sensors or target proteins as a next step. The group of target proteins includes a high variety of enzymes able to influence gene expression, metabolism, ROS detoxification, and protein turnover among other cellular processes [26]. Thiol switches (proteins characterized by redox-dependency of their catalytic or binding activity as based on cysteine residues) and thiol/disulfide exchange cascades are the main mechanistic elements of target proteins and the overall redox-regulatory network [2].

An example of a target protein is cyclophilin 20-3 (Cyp20-3), which contributes to thiol synthesis. Cyp20-3 is the only cyclophilin localized in the chloroplast stroma and acts as a regulatory hub between stress signaling of high light stress and wounding [27]. Under light conditions, Cyp20-3 is reduced by thioredoxins (TRX). In its reduced state, Cyp20-3 interacts with serine acetyl transferase 1 (SAT1), enabling formation of the cysteine synthase complex and subsequent production of the cysteine precursor *O*-acetyl serine [28,29,30]. Consequently, the cellular reduction state increases, changing the redox potential of the cell and allowing for alteration of gene expression and increased detoxification of ROS by peroxidases such as the ROS sensor 2-cysteine peroxiredoxin (2-CysPRX). Besides regulation of thiol synthesis, electrons from active Cyp20-3 can also be transferred to 2-CysPRX to enhance H_2_O_2_ detoxification [29].

#### 2.2.2. Exemplary Function of the Redox Network: The Water-Water-Cycle

An example of the interplay of ROS and the redox network is the plastid water-water-cycle, depicted in Figure 2. As previously mentioned, ROS generation in chloroplasts is especially high under non-optimal conditions due to univalent reduction of O_2_ by the redox input element ferredoxin in the course of the Mehler reaction [31,32]. The superoxide radical is rapidly converted into H_2_O_2_ by a (membrane-bound) Cu/Zn SOD, which acts as redox sensor [33,34]. Subsequently, H_2_O_2_ is detoxified by ascorbate peroxidases (APX) under consumption of ascorbate (ASC) [33,35]. The peroxidases, in this case, act as ROS scavenging enzymes; however, they additionally serve as redox sensors in plant cells [2]. Coupling of the water-water-cycle to the ascorbate-glutathione pathway (Asada-Halliwell pathway) ensures regeneration of oxidized dehydroascorbate [36]. Reduction of monodehydroascorbate (MDHA) occurs directly by ferredoxin oxidation or NADPH oxidation catalyzed by monodehydroascorbate reductase (MDHAR). Alternatively, MDHA can be further oxidized to didehydroascorbate (DHA), yielding one molecule of ascorbate. MDHA is then reduced to ASC by GSH-dependent dehydroascorbate reductase (DHAR); GSH, in turn, is regenerated by glutathione reductase (GR), using NADPH+ H^+^ [34].

### 2.3. Decomposition of ROS

Due to their high reactivity, ROS typically have short half-life times ranging from 10^−3^ to 10^−9^ s [5,6,9,15,16,24,37]. Although dismutation of ROS also occurs spontaneously, dedicated enzymes as catalysts (e.g., SODs) increase the reaction speed by a factor of 10^4^ [6,8]. Hence, detoxification of ROS by enzymatic antioxidants is achieved by a variety of stress-induced enzymes including SODs, catalases (CATs), and various peroxidases, including APXs, glutathione peroxidases (GPXs), glutathione-like peroxidases (GPXLs), and peroxiredoxins (PRX). Fine-tuning of ROS detoxification and, concomitantly, the ROS signature are realized by differential localization and the abundance of enzymes depending on developmental stage and tissue [6,8]. CATs, for instance, are thought to detoxify H_2_O_2_ in peroxisomes, but not in chloroplasts [6,8].

In addition, peroxidases serve as both a detoxification mechanism and as redox sensors; as such, they contribute distinctly to ROS signaling [2,4]. Key components of non-enzymatic ROS detoxification are GSH, ASC, carotenoids, tocopherols, and flavonoids [6,36]. As the primary ROS producer, chloroplasts are prone to oxidative damage. To counteract this threat, both enzymatic and non-enzymatic antioxidants accumulate in chloroplasts at high levels. Between 30% and 40% of the total cellular ASC content is stored in chloroplasts [6,11,38].

## 3. Oxylipins

### 3.1. Non-Enzymatic and Enzymatic Lipid Peroxidation Yields Highly Diverse Oxylipins

Oxylipins are bioactive lipid derivatives generated by oxidation of PUFAs that regulate plant growth, development, and stress defense [39,40,41,42]. While they are produced in all domains of life, species from each kingdom differ in the usage of PUFAs as a starting substrate: in plants, enzymatic oxylipin synthesis starts from α-linolenic acid (α-LeA) and linoleic acid, while animals commonly use arachidonic acid and eicosapentaenoic acid [40,43]. While the set of substrates for oxylipin synthesis is limited, a variety of products are generated by different pathways following PUFA oxidation (Figure 3).

In plants, oxygenation of α-LeA yields allene oxides and α-hydroxy PUFAs by the action of lipoxygenases (LOXs) and α-dioxygenases, respectively [40,41,44,45]. LOXs can be divided into two groups: 9-LOX and 13-LOX. Both groups catalyze oxygenation of PUFAs; however, their regiospecificity varies with 9-LOX introducing oxygen at the C9 carbon atom and 13-LOX at position C13 [40,44,46]. Hence, either (9S,10E,12E,15E)-9-hydroperoxyoctadeca-10,12,15-trienoic acid (9-HPOT) or (9Z,11E,13S,15Z)-13-hydroperoxyoctadeca-9,11,15-trienoic acid (13-HPOT) are generated and can be further metabolized by multiple enzymes to yield diverse oxylipins. A total of seven different pathways of 9-/13-HPOT derivatization has been described; a simplified overview of the synthetic pathways important for this review is provided in Figure 3 [41]. This review will focus (primarily) on oxylipins of three pathways: 12-OPDA as an example for the group of jasmonates, derived by the allene oxide synthase (AOS) pathway; 4-HNE and green leaf volatiles (GLVs) generated by the hydroperoxide lyase (HPL) pathway; and phytoprostanes, MDA, and acrolein as example of non-enzymatically derived oxylipins. 12-OPDA serves as both a signaling and regulator molecule on its own and as a precursor of JA. JA and its derivatives will not be included in this review, since extensive information on JA function can be found in recent reviews [47,48,49].

In addition to enzymatically formed oxylipins, a substantial amount of oxylipins is generated non-enzymatically by lipid peroxidation through free radicals [41,50]. Membrane lipids are the most prominent targets of free radicals in plant cells [51]; this is why lipid peroxidation serves as an early sign of oxidative stress. Important oxylipins of this pathway are phytoprostanes and their derivative (MDA), as well as acrolein [41,52,53]. Interestingly, the base concentration of diverse non-enzymatically formed oxylipins has been reported to be much higher than that of their enzyme-derived counterparts [54]. For instance, MDA levels surpass those of JA up to 50-fold [52].

Independent of their synthesis, oxylipins can be separated into two groups: reactive electrophile oxylipins (RES-oxylipins), also referred to as reactive carbonyl species (RCS), and non-reactive oxylipins [24,52,55]. Reactive electrophiles share a structural similarity, specifically the α, β-unsaturated carbonyl moiety, often as part of a cyclopentenone ring (Figure 4). This structure enables them to react with free thiol groups by Michael addition, regulating protein activity and the available pool of antioxidant molecules such as glutathione [27,56,57,58,59]. Hence, modification of thiol groups is a major mechanism of action shared by phytoprostanes, GLVs, non-GLV aldehydes, and JAs.

In general, RES can significantly damage plant cells and act as toxic and mutagenic agents [60]. For example, MDA damages DNA through reaction with guanine [53,60,61]. However, RES play important roles in signal transduction and defense mechanisms against biotic and abiotic stress; as such, they are regarded as a “REScue” mechanism of cells [55,60,62].

#### 3.1.1. Phytoprostanes Are Evolutionary Ancient Oxylipins

Phytoprostanes derive non-enzymatically from α-LeA and are precursors of MDA [51,52,63]. Through free radical activity, α-LeA is oxidized and a linolenate radical is generated. After additional autoxidation and cyclization, this radical forms phytoprostanes of two regioisomeric classes: the 9- and 16-series of phytoprostanes [51]. With a total of 32 isomers, the phytoprostane group contains a diverse set of oxylipins [63].

As they appear to have evolved distinctly earlier than other oxylipins and are abundant in most organisms, they are considered as an evolutionary ancient mechanism against oxidative stress [64]. Interestingly, cyclopentenone-containing phytoprostanes induce the expression of stress defense-related genes; these include glutathione-S-transferases (GSTs) which catalyze binding of RES to glutathione, one of the most abundant antioxidant molecules of plants [64,65]. Moreover, phytoprostanes stimulate production of phytoalexins, antimicrobial secondary metabolites, and thus enhance biotic stress tolerance [54,64]. This feature, as well as the regulation of gene expression and subsequent upregulation of GSTs, is not unique to phytoprostanes, but rather a common trait of oxylipins shared also by 12-OPDA, MDA, acrolein, and E-2-hexenal, the latter of which belongs to the group of GLVs [57,64,66,67,68,69,70,71].

#### 3.1.2. 12-OPDA and OPDAylation as Potent PTM

Cis-12-OPDA, which belongs to the group of jasmonates, is derived by the action of 13-LOX and the AOS pathway, which yields (9Z)-(13S-)-12,13-epoxyoctadeca-9,11,15-trienoate (12,13-EOT). As 12,13-EOT is highly unstable, it is rapidly converted to cis-(+)-12-OPDA by allene oxide cyclase (AOC), or, if there is no AOC available, to α- and γ-ketols and racemic 9R-13R-12-OPDA [27,56,72,73]. After enzymatic synthesis of 12-OPDA, it can either function as a potent signaling molecule and phytohormone or be transported to the peroxisomes, where it is used to generate JA [27,47,74]. JA, in turn, is further modified, e.g., by conjugation to amino acids such as isoleucine (yielding JA-Ile), as catalyzed by JASMONIC ACID RESISTANT 1 (JAR1) [47,75]. JA-Ile and cis-12-OPDA are regarded as the biologically active forms of JA and 12-OPDA, respectively [76,77]. Recently, conversion of 12-OPDA to dinor-OPDA (dn-OPDA), a homologue of OPDA with an acyl residue shortened by two carbons, usually synthesized from hexadecatrienoic acid, was detected in *Arabidopsis thaliana* [78,79]. An overview of OPDA synthesis and metabolic conversion is provided in Figure 5.

Under stress conditions such as wounding, high light or pathogen attack, 12-OPDA synthesis is highly stimulated [27]. Using transgenic *A. thaliana* lines which express the Pseudomonas syringae avirulence peptide AvrRpm1 depending on dexamethasone, Andersson et al. observed an increase of 12-OPDA from basal concentration of 2 µM to 36 µM after 4 h of AvrRpm1 expression. Interestingly, accumulation of lipid-bound 12-OPDA, so-called Arabidopsides was even more pronounced with a 200-fold increase [42].

Arabidopsides are cyclo-oxylipin-galactolipids in which 12-OPDA is bound to complex membrane lipids such as mono- and digalactosyldiacylglycerol (MGDG/DGDG) [40,41,80,81]. They are unique to certain species, primarily of the Brassicaceae family (e.g., *A. thaliana*, *Camelina microcarpa*, *Capsella rubella*), but also Poaceae (*Hordeum vulgare*), Convolvulaceae (*Ipomoea tricolor*), and Lamiaceae (*Melissa officinalis*) [39,40,76,80,81,82,83,84]. Concerning synthesis of Arabidopsides, incorporation of free 12-OPDA to membrane lipids as well as direct membrane-bound generation from α-linolenic acid are discussed, with the existence of membrane-bound 13-HPOT in Arabidopsis arguing for the latter [42,84,85,86]. While increases in Arabidopside content due to unfavorable environmental conditions could be observed in several studies, their function in plants still remains mostly unknown [42,80,82,85,87]. On the one hand, they have been proposed to serve as storage pools for the fast release of 12-OPDA under stress, either as a means of rapidly increasing OPDA signaling or increasing synthesis to JA [42,82,86,87]. Several enzymes have been linked to this release of 12-OPDA, including acylhydrolase, phospholipase 1 (PLAI), DEFECTIVE IN ANTHER DEHISCENCE1 (DAD1), and DONGLE (DGL) [27,88,89]. On the other hand, Arabidopside E and G possess antimicrobial functions; Arabidopside A stimulates senescence in barley leaves and Arabidopside A and B inhibit root growth, indicating the possibility that Arabidopsides themselves are functional molecules of stress signaling and defense [42,86,90]. Furthermore, Arabidopsides might interact with glycosyl inositol phosphor ceramides, i.e., major sphingolipids in plants linked to stress defense [91,92].

Similar to phytoprostanes, jasmonates play an essential role in plant defense against different abiotic and biotic stresses [27,93,94]. Additionally, they influence plant growth and reproduction. For instance, jasmonates regulate pollen maturation, elongation of stamen filament, and pollen release as well as leaf movement and fruit ripening [76,93,94,95]. Further overviews on the role of 12-OPDA in plants can be found in recent reviews [27,96,97].

#### 3.1.3. Oxylipin Aldehydes

Similar to ROS, high concentrations of oxylipin aldehydes can lead to extensive cell damage, while lower concentrations contribute to cell signaling processes in the context of abiotic stress [98,99,100]. In general, lipid-peroxide derived aldehydes inhibit seed germination, CO_2_ photoreduction, and plant growth and senescence [98,101,102].

Unlike the oxylipins mentioned so far, 4-hydroxy-2-nonenal (4-HNE) is a lipid aldehyde that derives from linoleic acid by action of 9-LOX, hydroperoxide lyase (HPL), an alkenal oxygenase, and hydroperoxide peroxygenase [103]. 4-HNE has been regarded as the most toxic and most abundant product of lipid peroxidation and primarily accumulates in biomembranes [103]. Besides thiol binding by Michael addition, binding of 4-HNE to Arg residues leading to 2-pentylpyrrole adducts has been observed [104]. As the latter has only been documented in animals, a similar effect of 4-HNE remains to be proven in planta.

MDA is the smallest molecule discussed in this review and only consists of three carbon molecules and two aldehyde groups [105]. Nevertheless, it is highly reactive and commonly regarded as biomarker of oxidative stress [105]. For instance, accumulation of MDA under heat stress negatively correlates with functionality of the photosynthetic electron transport chain [99,105]. However, as with ROS and oxylipin aldehydes in general, MDA fulfills a dual role in plants, acting as signaling molecule in stress defense and acclimation processes as long as its concentration is correctly balanced [105].

The C6 aldehyde 2-hexenal belongs to the subgroup of GLVs and the bigger group of biogenic volatile organic compounds (BVOCs) [41,106]. Emission of VOCs accounts for up to 10% of total fixated carbon in plants with 10^9^ tons VOCs per year [107,108]. Production and release of VOCs is tightly regulated by biotic factors (pollination status, herbivore infestation) and abiotic factors (light intensity, atmospheric CO_2_, temperature, humidity, nutrition) in a spatial, temporal, development-specific, and species-dependent manner [107,108]. To be released, VOCs must cross membranes, the cell wall, and (depending on the tissue) also the cuticle, a major barrier in most plant cells [107,108].

Due to their high lipophilicity, as evident from their high octanol-water partition coefficient, VOCs are primarily found in hydrophobic environments where they can significantly damage cellular structures [107,108]. To ameliorate this damaging effect, the cuticle serves as a volatile sink and storage pool, primarily for VOCs with low volatility [108]. As a consequence, VOC emission depends on the composition and thickness of the cuticle [107].

### 3.2. Oxylipin Signature

The oxylipin composition of cells and tissues, the oxylipin signature, varies depending on plant species, organ and tissue, developmental stage, and environmental conditions [109,110,111]. By fine-tuning oxylipin concentrations, different signaling pathways and plant processes can be targeted. An example of a species-dependent oxylipin signature is the induction of phytoalexin synthesis. In rice and tobacco, both JA and 12-OPDA stimulate phytoalexin production in leaves (as measured as sakuranetin and scopoletin contents); however, in soybean, only OPDA can activate phytoalexin synthesis [112,113,114]. Species-dependent oxylipin signatures can be traced back to both differing oxylipin functions and to differences in the oxylipin concentrations needed for their functionality. For instance, high concentrations of jasmonates could be measured in sorbitol-treated *Hordeum vulgare* with 5 nmol g^−1^ FW of 12-OPDA (1462 ng g^−1^ FW) and 2.2 nmol g^−1^ FW of JA (462 ng g^−1^ FW) after 24 h [115]. In contrast, osmotic stress caused by sorbitol treatment of *A. thaliana* for the same time span results in a lower accumulation of 12-OPDA and JA with 462 and 256 ng g^−1^ FW, respectively [116]. As sorbitol was employed at twice the concentration in the first study when compared to the second study (1 M and 0.5 M, respectively), one might argue that OPDA accumulates significantly due to enhanced osmotic stress caused by high sorbitol concentrations. Nevertheless, dependency of oxylipin composition on plant species should be considered in this case as well.

Interestingly, the type of attacking insects also influences the oxylipin signature under biotic stress. For instance, piercing-sucking and chewing insects both stimulate emission of LOX-derived volatiles in distinctly different degrees. After aphid attack, LOX-derived volatiles accounted for 8.9% of total volatiles; chewing insects, meanwhile, caused an increase of this percentage to 53%. Additionally, the composition of volatiles varies, with C6-volatiles being more strongly emitted after aphid infection and C9-aldehydes (as measured on nonanal) after infestation with chewing insects [117].

Consequently, regulation of oxylipin signatures is highly complex and more detailed studies, concentrating not only on single oxylipins but on the broad profile simultaneously, are needed to understand how oxylipins affect plant processes [50].

Modulation of oxylipin composition might be achieved by differences in the ROS level (concerning non-enzymatically derived oxylipins) and regulation of the activity of different enzymes such as LOXs. For example, if 13-LOX is up- and 9-LOX downregulated, the ratio of 13-HPOT to 9-HPOT increases. The changed ratio results in enhanced synthesis of jasmonates relative to GLVs.

## 4. Influence of Oxylipins on the Redox-Regulatory Network

### 4.1. Modulation of Thiol-Sensitive Proteins

Oxylipin RES impact the redox regulatory network at all levels, from redox input elements to redox sensors and target proteins, by influencing gene expression, protein synthesis, and catalytic activity. Using an anti-HNE antibody, Mano et al. described a subset of 34 proteins modified by, as they assumed, by both 4-HNE RCS in general. This subset contains essential proteins of ROS generation and the redox network such as peroxidase 34, a cell wall peroxidase linked to oxidative burst, or Cyp20-3 and cysteine synthase, both dedicated to thiol synthesis [118].

Synthesis of heat shock proteins, stress-induced chaperones, is upregulated not only by 12-OPDA and phytoprostanes but also by MDA [66,119], while, after synthesis, they are targets of 4-HNE [103]. Other universal targets of RCS include GSTs, as mentioned earlier, and, interestingly, enzymes usually linked to carbon metabolism, more specifically glycolysis. NAD-dependent glyceraldehyde-3-phosphate dehydrogenase (GAPDH) is not only dedicated to glycolysis but also exerts moonlighting functions including autophagy, apoptosis, and translation by RNA binding [120,121,122]. As its function strongly depends on its cysteinyl residue redox state, GAPDH belongs to the group of target proteins [123,124]. Oxylipins play a dual role in altering GAPDH levels and activity. Covalent binding of 4-HNE to GAPDH and the subsequent inhibition of catalytic activity has already been shown by Uchida and Stadman in 1993 [125]. Binding of acrolein also decreases GAPDH activity [102,126]. Moreover, binding of 4-HNE as well as 4-HHE appears to trigger GAPDH degradation [127].

On the other hand, a stimulatory effect of 12-OPDA on GAPDH expression has been proposed based on studies with *Physcomitrella patens* mutant lines lacking AOS activity and 12-OPDA-treated *Pohlia nutans* [128,129]. Assessment of differentially expressed genes due to 12-OPDA treatment in various organisms (*P. nutans*, variegated *Epipremnum aureum*, and the algae *Klebsormidium nitens*) shows upregulation of gene orthologues of *A. thaliana* GAPDH isoforms (GAPCp1, GAPC2), whereas GAPA2 and GAPB orthologues are downregulated [79,129,130]. The only direct study of 12-OPDA-responsive genes (ORGs) in *A. thaliana*, conducted by Taki et al., does not show significant regulation of GAPDH expression by 12-OPDA. However, although not fulfilling the criteria for ORGs (which were quite stringent, with a minimum of 3-fold relative expression increase in this study), gene expression of GAPC1 shows a trend towards upregulation with an approximately twofold increase [66]. Altogether, one might tentatively expect a stimulatory effect of 12-OPDA on GAPDH.

In vitro, OPDAylation of GAPDH (as tested using GAPC2) only showed minor inhibition on its NADH oxidation activity under physiological 12-OPDA concentrations [56]. Modulation of GAPDH signaling by oxylipins might influence carbon metabolism and the energetic state of the cell. Stimulation of GAPDH levels by 12-OPDA could serve as a means of counteracting energy consumption due to stress defense mechanisms. On the other hand, GAPDH also exerts RNA binding functions, hence modulating protein synthesis [120]. RNA binding motifs of GAPDH are primarily found in high light-induced transcripts [131]. Therefore, modulation of GAPDH levels and oxidation state might help in steering protein synthesis in the direction of stress signaling and defense, similar to the induction of ORGs by interaction with Cyp20-3. Similar to GAPDH, aldehydes such as acrolein target and inactivate fructose-bisphosphate aldolase (FBPase), another glycolytic enzyme and redox target protein for which RNA binding activity has been described [98,102,120]. However, in plants, only nonspecific binding of plastid FBA1 to RNA coding for a subunit of the cytochrome b6/f complex has been shown [132].

Modulation of Cyp20-3 activity by oxylipins, more specifically non-covalent binding of 12-OPDA, has been shown to enhance Cyp20-3 activity. Consequently, thiol synthesis is stimulated, strengthening the redox capacity of the cell. This change in redox state in turn alters the expression of 12-OPDA-responsive genes [27,28,96,133]. Contrary to Cyp20-3 stimulation, OPDAylation diminishes the H_2_O_2_ scavenging function of 2-CysPRX [133]. OPDAylation of TRXs drastically impedes their activity as redox transmitters, thereby decreasing the rate of TRX-dependent Cyp20-3 reduction and glutathione peroxidase (GPXL) regeneration. The overall functionality of this inhibition might be explained by different modes. First, inhibition of TRX activity and subsequently increased ROS productions might enhance ROS-induced stress signaling [56,134,135]. Second, inhibition of chloroplast TRXs also diminishes TRX-dependent FBPase reduction. As a consequence, CO_2_ assimilation and starch synthesis are inhibited, reducing the energy consumption by the Calvin–Benson cycle [133].

SODs are additional targets of oxylipins. While 4-HNE has been shown to bind to Mn SODs, 12-OPDA downregulates gene expression of Cu/Zn SODs; these are part of the water-water cycle [103,136]. Since acrolein releases Zn^2+^ from different proteins, an inhibitory effect of this RCS on Cu/Zn SODs followed by upregulation of ROS levels has been proposed [100,137].

Exogenously supplied acrolein, 4-HNE, and HHE increase catalase and APX catalytic activity in *A. thaliana* [138]. Moreover, 4-HNE interacts with MDHAR. An oxidizing effect might be assumed, but remains to be proven, whereas 12-OPDA enhances DHAR synthesis [103,139]. An exemplary overview of oxylipin-dependent regulation of redox network proteins can be found in Table 1.

### 4.2. Interaction with Non-Protein Thiols

Non-protein targets of oxylipins include ASC and GSH. Both compounds constitute abundant cellular redox buffers that are essential, e.g., in the Asada–Halliwell pathway [27,36]. Glutathionylation of RES, including cyclopentenone phytoprostanes, inhibits their activity and thus protects cells from damage such as the inactivation of thiol switch proteins [65,68,140]. Regarding the main oxylipins included in this review, only 4-HNE, acrolein, and 12-OPDA have been shown to undergo GSH binding so far (either spontaneously or catalyzed by GST6, respectively) [70,119,141]. However, the fate of these adducts remains mostly unknown; it is expected that adducts are transported to the vacuole for degradation, as is also the case for GSH-OPDA adducts [67].

Possible new functions of the GS-adducts in vivo await elucidation. In animals, studies indicate a unique ability of GS-HNE in stress defense modulation when compared to free 4-HNE, with a function in regulating the transcription factor NF-κB [142]. Concerning cyclopentenone prostaglandins PGA2 and PGJ2, GSH binding serves as a “shuttle”, enabling transport of otherwise highly hydrophobic molecules into different cell organelles. After reaching their target destination, PGs might be released from GSH (either spontaneously or enzymatically catalyzed) by undergoing retro-Michael Addition and binding to higher affinity protein thiols [143]. This mechanism, based on increasing hydrophilicity after GSH binding, is likely transferable to plant oxylipins and broadens the spectrum of cell organelles containing proteins sensitive to OPDAylation and OPDA-dependent regulation.

To this day, the only effect of GS-OPDA adduct formation concerns the 12-OPDA localization by import into the vacuole, where it undergoes degradation [67]. Hence, this might be considered as a purely protective function to remove this compound from the cytoplasm. However, this process should be studied more extensively in order to clarify whether GSH binding has additional functions in cellular regulation.

GSH concentrations surpass those of typical oxylipins several-fold. GSH concentrations in plasmatic compartments are in the mM range, whereas oxylipins, including 12-OPDA, reach µM values. Given these molar ratios, the question of how free oxylipin functions are maintained despite their reactivity with GSH must be discussed. First, the reaction kinetics between oxylipins and proteins on the one hand and oxylipins and GSH on the other hand differ greatly in dependence on the pK_A_ values of the Cys residues. Acrolein binds rapidly to protein thiols; however, when compared to HHE and 4-HNE, GSH-acrolein adducts form 70- to 110-times faster, respectively [53,100,126]. For example, studies on the interaction of 4-HNE and GSH revealed a half-life time of 2 min for 4-HNE in the presence of 5 mM GSH [144]. Second, adduct formation of GSH and oxylipins is a reversible process that includes characteristic adduct stabilities for each oxylipin. For instance, whereas acrolein-GSH adducts are very stable with a half-life time of 4.6 days, hexenal-GSH adducts decayed with a half-life time of 4 to 6.3 h [145].

Further, oxylipins influence the cellular GSH pool and synthesis and activity of GSTs. *Trans*-2-hexenal inhibits pumpkin GST, though only at high concentrations (mM range) [70]. Aldehyde dehydrogenases (ALDHs) function as important enzymes by maintaining the glutathione pool in a reduced state and detoxifying RCS. ALDHs are targets of 4-HNE [101,103,146]. Depletion of chloroplast GSH content is a common feature of lipid-peroxide derived aldehydes [98].

To conclude, oxylipins commonly decrease cellular GSH levels. High concentrations of acrolein even deplete the cell of GSH and ascorbate. This might be due to inhibition of GSH synthesis as well as stimulation of GSTs and binding of GSH to the oxylipins themselves. Contrary to this, 12-OPDA influences GSH levels positively by enhancing Cyp20-3-mediated thiol synthesis. Whether oxylipins are completely inactivated after binding to GSH and whether adducts gain additional functions or only serve as shuttles remains to be elucidated.

### 4.3. Contribution of Oxylipins to Environmental Acclimatization

Due to climate change and global warming, plants consistently face changing environmental conditions and extended periods of severe biotic and abiotic stress. On the one hand, species dominance shifts by changes in land use, and more extensive outbreaks of insects are expected, especially in forests [106,147,148]. Simultaneously, temperature variation becomes more extreme with the mean global temperature rising; however, more pronounced cold periods may occur, while the frequency of flooding and drought events rises [106,147,148,149]. Not only is soil nutrient availability changing, but the pollution of soil (as well as air and water) is constantly increasing as a result of anthropogenic activities [106,149].

All these stresses threaten yield formation as plants invest more resources in defence as a trade-off to survive. For instance, under UV stress, plants might encounter morphological changes and altered genome stability due to DNA damage [108]. Oxylipins and redox state are major players in the plant-environment interaction, e.g., the adaptive mechanisms include upregulation of cuticle synthesis. This upregulation, in turn, changes VOC emission, their distribution in planta, and their biosynthesis [108]. Similar effects on cuticle synthesis can be observed under drought stress, flooding stress, high salinity, and high and low temperature [150,151]. Abiotic stress, such as intense heat or drought periods, and severe climatic events, such as flooding, require a constant evolutionary adaption and flexible acclimatization of plants to their environment.

#### 4.3.1. Thermotolerance

Changing temperature and air pollution disturbs signalling through GLVs. Air pollution results in a decrease of maximum downwind distance of reactive volatiles, while cold temperatures cause GLVs to condense and reside on plant surfaces instead of emitting into the atmosphere [106,147]. Hence, pollination of plants deteriorates as the searching and foraging efficiency of pollinators decreases. At the same time, the diminished attraction of predators prevents anti-herbivore defence and might lead to more extensive wounding of plants with increased synthesis of oxylipins. This adds to a generally increased severity of biotic stress, which is even further pronounced due to the rising severity of insect outbreaks [147].

On the other hand, emission of BVOCs, including GLVs, has already increased by 10% over the last three decades due to climate change; this is estimated to increase by a further 30–45% as global temperature rises by another 2–3 °C [106]. This increase might be traced back to indirect and direct effects on BVOC emission, which is characterized not only by their generation, but also by their physicochemical properties (solubility, volatility, diffusivity). Emission rates are affected indirectly by prolongation of plant growth phases and directly by affecting biochemical generation of BVOCs, both due to increasing temperature [106]. Considering high temperatures as the only factor, emission of BVOCs is proposed to be more potent, as their vapour pressure and their volatility enhance their diffusivity at the same time [106]. However, as mentioned before, cold temperatures and air pollution might counteract these effects.

RES oxylipins are proposed to act as an ancient system of stress defense regarding thermotolerance, which evolved distinctly before JA signaling [79]. To counteract heat stress, 12-OPDA, phytoprostanes, and MDA induce synthesis of heat shock proteins, major chaperones involved in heat stress acclimation [66,96,119].

As with heat stress, cold stress severely limits plant performance and may ultimately lead to plant death in non-adapted species. Tolerance to both stresses, moreover, involves JA and SA signalling [152]. In rice, under cold stress, several genes of jasmonate biosynthesis are upregulated, including OsLOX2, OsAOS1, and OsOPR1, resulting in accumulation of JA (and possibly also 12-OPDA) [152]. JA signalling increases plant endurance and freezing tolerance [152,153]. However, as shown for *A. thaliana*, this upregulation of jasmonates appears to be limited to the first exposure to cold stress and dramatically decreases in primed plants [154]. This might be explained as a protection mechanism to reduce costs for gene expression of cold-tolerance genes under fluctuating weather conditions, for example in spring [154].

#### 4.3.2. Pathogen Infection and Induced Systemic Resistance (ISR)

Besides abiotic stress, biotic stress, as caused by pathogen infection, insect attack, or intraspecific competition, significantly impacts pre- and postharvest crop yield [155,156]. In regards to most prominent food crops, including wheat, rice, maize, and potato, biotic stress has been reported to account for roughly 28%, 37%, 31%, and 40% of yield losses from 2001 to 2003, respectively [157,158]. In general, biotic stress is thought to cause over 40% of yield loss in global food production [157,159,160].

As recently reviewed, JA and its derivate JA-Ile are major regulators of herbivory- and pathogen-induced defence mechanisms in plants [161,162]. However, other oxylipins, besides JA and JA-Ile, further contribute to herbivory resistance. In organisms devoid of JA signalling, such as *Marchantia polymorpha*, dinor-OPDA, instead of JA, mediates herbivory defence signalling together with salicylic acid (SA) [163]. Moreover, 12-OPDA, possibly in concert with dn-OPDA, maintains plant resistance against insects and fungi in absence of JA in *A. thaliana* [164].

Feeding preferences of herbivores depend on *cis*-3-hexenal, a GLV that stimulates feeding activity [165]. In *A. thaliana*, signalling from wounded shoot tissue to distal parts of the plant depends on the transport of 12-OPDA through the phloem from shoot-to-root, where it then mediates activation of JA and JA-Ile signalling [166]. GLVs are important signalling molecules involved in intra- and interspecies communication, even across kingdoms. For instance, after herbivorous damage, there exist a means of attracting predators of herbivorous insects and enhancing plant defence against pathogens and nematodes [68]. Further, they serve as warning molecules for neighbouring plants. GLVs that are emitted from plants subjected to biotic stress can be perceived by surrounding plants. This induces stress defence mechanisms or primes the plant, resulting in a pre-defence state in which stress defence induction proceeds more quickly and strongly [68,167]. The mechanism of GLV perception in plants, however, is still largely unknown [68]. Lastly, GLVs are essential molecules for attracting pollinators and, thereby, maintain plant reproduction [68,106,168,169].

Oxylipins (phytoprostanes, OPDA) distinctly enhance phytoalexin synthesis [54]. Furthermore, due to their chemical structure (dependent on carbon chain length, double bonds, and hydroperoxyl groups), oxylipins display antimicrobial effects. They cause cell lysis and subsequent electrolyte leakage in gram-positive and gram-negative bacterial cells and inhibit enzymes, including those dedicated to nutrient uptake, ATP synthesis, and the respiratory ETC [170,171,172,173,174,175,176,177].

Symbiosis of plant roots and rhizobacteria not only benefit for plant growth but can also trigger induced systemic resistance (ISR). ISR is a type of defence mechanism against pathogens and herbivores that relies on signalling molecules transported from roots to aboveground plant tissues (root-to-shoot signal molecules) [178,179]. In maize roots colonized by *Trichoderma virens*, ISR is activated by fungal secretion of oxylipin synthesis inhibitors.

While Sm1, a hydrophobin-like elicitor, is secreted by *T. virens*, the production of 9-HPOT by LOX3 is inhibited. Although seven 9-LOX are produced in maize, LOX3 appears to be the only LOX responsible for regulation of ISR, as LOX3 knockout-mutants show constitutive ISR [180,181,182]. However, plant lines deficient in LOX5 and LOX10 also display enhanced resistance to parasite infection, which raises the question of whether they might be inhibited in the absence of LOX3 activity [180,183,184]. Recently, 12-OPDA and α-ketol-octadecadienoic acid (KODA) have been identified as root-to-shoot signals in *Zea mays* [185]. Both depend on the AOS pathway of oxylipin synthesis; however, 12-OPDA is synthesized from 13-HPOT and KODA from 9-HPOT. As a result, activity of both 9- and 13-LOX is required. This might appear contradictory with the involvement of LOX3 in ISR. However, maize plants deficient in LOX3 activity characteristically overexpress genes of jasmonate biosynthesis such as LOX10 [186]. Production of KODA in maize has not been traced back to a specific 9-LOX so far, excluding LOX3, so KODA can still by synthesized by the other six 9-LOX.

All in all, while JA is the primary oxylipin involved in herbivory, the oxylipins that we focus on in this review fulfil additional and essential functions, such as root-to-shoot signalling and ISR. Furthermore, they contribute to pathogen defence and are able to substitute JA signalling in the absence of JA (depicted in Figure 6), as indicated by studies in different species.

Another important plant stress defense mechanisms against pathogen infection is hypersensitive response (HR): local necrosis, which prohibits further spread of infection to healthy tissue [42,187,188]. HR is driven by recognition of small elicitor molecules containing specific patterns, so-called pathogen-associated molecular patterns (PAMPs), by dedicated receptors (pattern recognition receptors, PRRs) as well as detection of avirulence (avr) proteins by nucleotide-binding and leucine-rich repeat receptors (NLRs) and resistance (R)-proteins [187,189,190,191,192]. After recognition of PAMPs or avr proteins, a signalling cascade involving NADPH oxidases (RBOH) is initiated that causes an oxidative burst and subsequently programmed cell death (PCD) [42,193,194].

As regulation of HR constitutes a complex network of transcriptional, post-transcriptional, and post-translational modifications, this review will only provide limited insight as related to oxylipin-linked HR. While concrete mechanisms and redox-regulatory pathways behind HR-associated PCD still await further elucidation, the overall state of current knowledge on HR has been focus of recent reviews [187,192,195,196]. One protein involved in post-translational modification of HR is HSP90, which, together with additional chaperones, stabilizes NLRs [187,197]. Moreover, disturbance of APX and MDHAR levels and activities has been linked to generation of the oxidative burst of PCD [198,199]. Consequently, as RBOHS, MDHAR, APX, and HSP90 have been shown to be affected by oxylipins, as described in Section 4.1 and Section 4.2, they might contribute possible links to oxylipin-modulated HR. As TRXs negatively regulate HR-driven PCD, inhibition of TRX activity by 12-OPDA might contribute to the rapid induction of PCD under stress conditions, as 12-OPDA synthesis is stimulated in the course of HR [27,200].

Another possible interaction node of HR signaling are ceramides, which are hypothesized to interact with Arabidopsides as well as the phytoalexin pool [201]. Moreover, Arabidopsides, specifically Arabidopside E, are thought to serve as protection against secondary infections in already dead tissue after PCD [42]. Although studies on the effect of oxylipins on HR are scarce, oxylipins have been proposed as determinants of HR [201]. For instance, stimulation of ROS-triggered programmed cell death (PCD) by oxylipins (acrolein, 4-HNE) after pathogen infection has been linked to depletion of the glutathione pool and lethal depletion of ASC, explained by stimulation of caspase-like proteases [202]. Negative regulation of HR by oxylipins has also been observed in *A. thaliana* cells treated with phytoprostanes, possibly due to activation and increased synthesis of GSTs, MAPK, and antimicrobial compounds [201,203]. Overall, both the mechanisms of HR-induced PCD and the possible involvement of oxylipins should be studied more extensively.

#### 4.3.3. Flooding

Another effect of climate change is more frequent (temporary) flooding, causing not only submergence (hypoxia) stress, but reoxygenation stress as an additional challenge. During submergence, oxygen availability decreases drastically, the composition of soil and accessibility of nutrients and microbial environment changes, and toxic compounds are formed [204]. Consequently, the photosynthesis rate of plants decreases as well as their energy and carbon levels, resulting in growth retardation. Temporary flooding is followed by reoxygenation, which leads to distinct oxidative stress in plants [204,205,206].

Oxylipins, especially 12-OPDA and C6 aldehydes of the group of GLVs, ameliorate flooding stress in plants and increase plant survival as depicted in Figure 7 [204,207]. By using different *A. thaliana* wild type and transgenic lines, Savchenko et al. analysed the function of the HPL and the AOS pathway, both dependent on 13-HPOT [208]. They showed a protective effect under waterlogging stress concerning biomass accumulation, membrane integrity, photosynthesis rate, and overall submergence survival rate. This effect relies not only on the combined action of both pathways, but also on the unique action of products of the HPL or AOS branch. For instance, the AOS branch appeared to be more critical for defence against membrane damage during reoxygenation as measured by electrolyte leakage and lipid peroxidation. However, in both cases, the combination of HPL and AOS activity resulted in the least damaged membrane.

Furthermore, the decline in photosystem II activity was less pronounced in the presence of oxylipins. This is in accordance with previous studies that show a protective effect of HPL on the photosynthetic apparatus under high light stress [209]. Overall, the survival rate of plants under submergence stress increases from ~50% when only one of the two pathways is active to ~77% with both pathways functional. In a double knockout mutant displaying neither AOS nor HPL activity, survival rate drops to ~30%, supporting the importance of oxylipins under flooding stress [204].

In plants displaying HPL and AOS activity, 12-OPDA accumulated after submergence stress. Since earlier studies reported transgenic *A. thaliana* lines deficient in JA biosynthesis and/or signalling to show increased sensitivity towards reoxygenation, and no increase of JA could be detected in this study, an essential role of 12-OPDA in flooding tolerance can be concluded. 12-OPDA could either be the only AOS-derived key regulator, or act together with basal concentrations of JA [204]. Additionally, 12-OPDA accumulates under combined flooding and heavy metal stress and has been proposed to enhance stress defense by upregulation of thiol production and tuning gene expression under these stress conditions [207].

In total, an essential role in plant survival under flooding stress can be attributed to oxylipins with 12-OPDA acting as a primary signalling molecule.

## 5. Conclusions

Under non-optimal growth conditions and severe stress such as flooding and heat, which are expected to further increase due to global warming over the next decades, plants are faced with an accumulation of ROS. As these can cause extensive damage, a complex array of proteins, the redox-regulatory network, is dedicated to maintaining ROS homeostasis. Products of ROS-induced damage, oxylipins, and the function of the redox network are strongly linked. This interplay awaits further elucidation, as studies on oxylipins are generally rather scarce, especially concerning studies on oxylipin signature instead of single oxylipins. Nevertheless, the strong effect of oxylipins on redox homeostasis in plant cells has been widely recognized; this interference affects plant performance in a larger context and in environmental acclimatization. The interaction between glutathione and oxylipins, especially in the context of adduct formation, function, and cleavage, should be further characterized, particularly concerning 12-OPDA, a precursor of jasmonic acid, that, in recent years, has emerged as a potent phytohormone on its own.

## Figures and Tables

**Figure 1 antioxidants-12-00814-f001:**
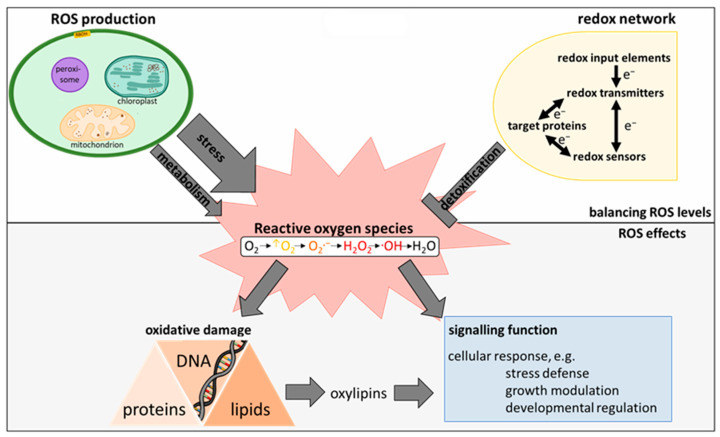
Regulation of ROS levels and their effect on plant cells. ROS are generated in cytosol, apoplast, plastids, mitochondria, and peroxisomes. ROS generation is strongly stimulated under stress conditions. Contrarily, decomposition of ROS is achieved by enzymatic and non-enzymatic antioxidants under control of the redox-regulatory network. As ROS inflict oxidative damage on proteins, DNA, and lipids (yielding e.g., oxylipins) while also fulfilling signaling functions, tight control of synthesis and degradation of ROS is essential for plant fitness and survival.

**Figure 2 antioxidants-12-00814-f002:**
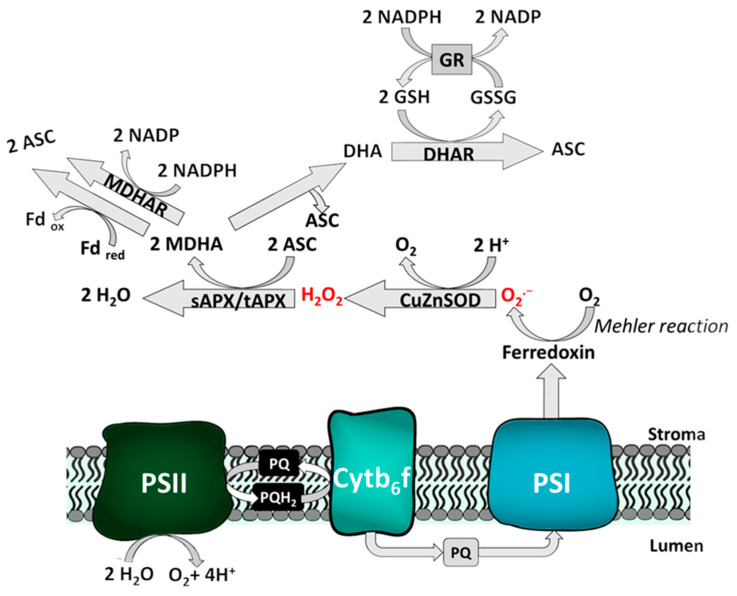
Generation and scavenging of ROS in the water-water-cycle. Due to over-reduction of the photosynthetic electron transport chain, super oxide radical is produced during the Mehler reaction. A set of enzymes and antioxidants is dedicated to detoxification of O_2_^•−^, followed by regeneration of oxidized ascorbate and GSH. See text for further details.

**Figure 3 antioxidants-12-00814-f003:**
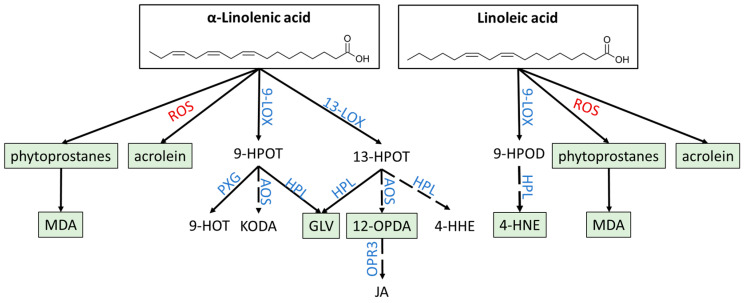
Synthesis pathways of oxylipins. Starting from oxygenation of α-linolenic acid or linoleic acid by LOX or reactive oxygen species, a variety of enzymes generates diverse oxylipins. Most important oxylipins covered in this review are marked in green. Blue: catalyzing enzymes; red: non-enzymatic conversion by ROS; dashed arrows: conversions that occur in multiple steps.

**Figure 4 antioxidants-12-00814-f004:**
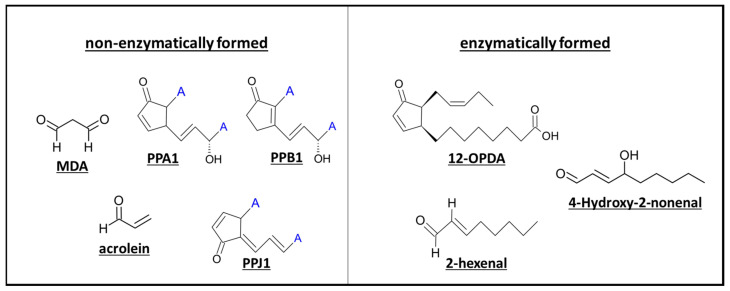
Structure of different α, β-unsaturated carbonyl oxylipins. Oxylipins of different synthesis pathways (phytoprostanes (PP), MDA, acrolein (non-enzymatic synthesis), 12-OPDA (AOS pathway), and 2-hexenal, 4-hydroxy-nonenal (HPL pathway)) contain a reactive α,β-unsaturated carbonyl compound and can thus participate in Michael addition.

**Figure 5 antioxidants-12-00814-f005:**
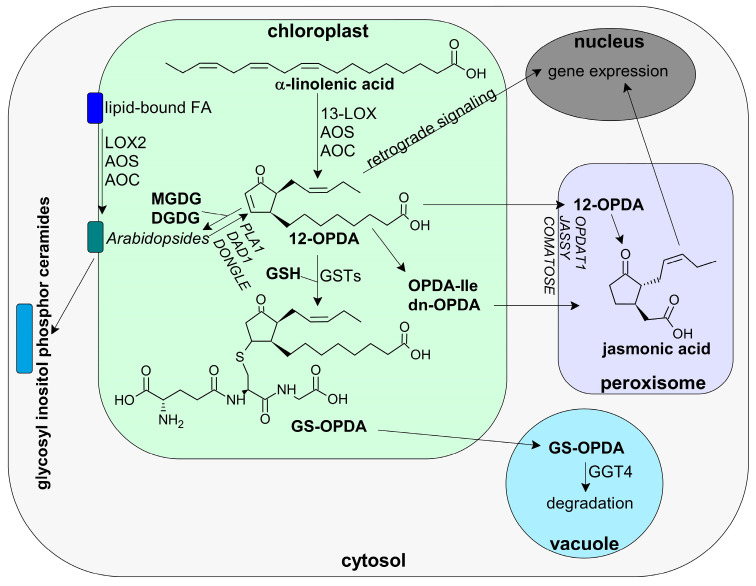
Synthesis and metabolic conversion of 12-OPDA in plant cells. After synthesis from α-LeA by 13-LOX, AOS, and AOC, 12-OPDA can be transported to peroxisomes through the cytosol and converted to JA, which regulates gene expression in a COI-1 dependent matter. Further, 12-OPDA can be stored by binding to MGDG and DGDG, forming Arabidopsides. These storage pools can also be formed directly from lipid-bound fatty acids (FA) by LOX2, AOS, and AOC and might interact with the major sphingolipids GIPC in plant membranes. Further, 12-OPDA forms adducts with GSH which are degraded in vacuoles by γ-glutamyl transpeptidase 4 (GGT4). Without further conversion, 12-OPDA serves as regulator of retrograde signaling.

**Figure 6 antioxidants-12-00814-f006:**
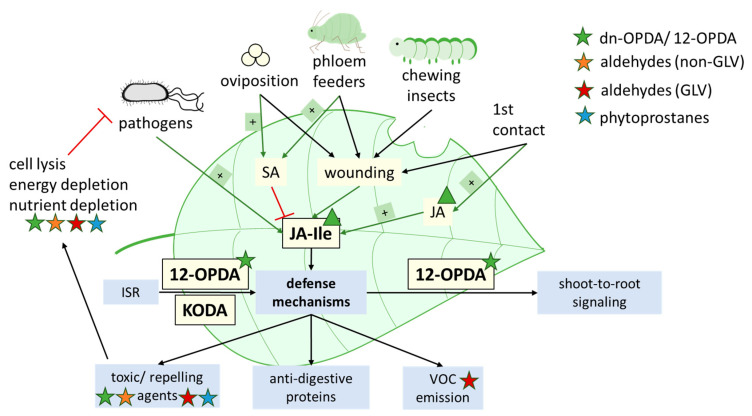
Oxylipins as main mediators of biotic stress (alongside SA). Besides JA, 12-OPDA, (non-GLV)-aldehydes, and phytoprostanes influence plant defense against pathogens and herbivory. Processes influenced by these oxylipins are indicated by colored stars. Functions fulfilled by (dn-) OPDA in the absence of JA are marked by triangles.

**Figure 7 antioxidants-12-00814-f007:**
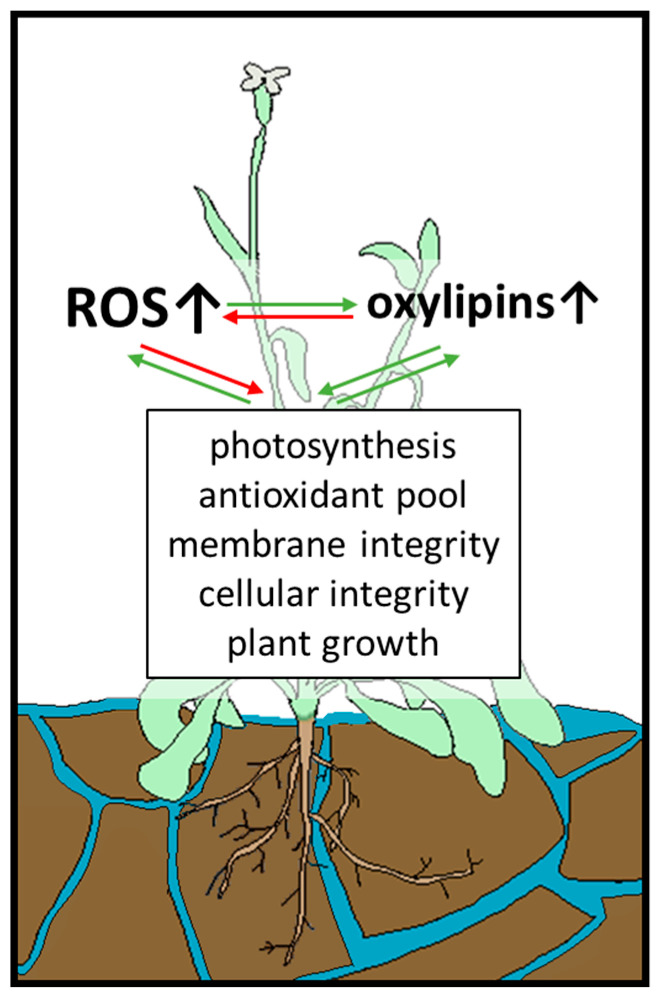
Waterlogging stress increases ROS content in plant cells, which impairs essential functions such as photosynthesis, and is counteracted by oxylipins. ROS accumulation can lead to extensive damage in plant cells, damaging, for example, membrane integrity, which leads to enhanced oxylipin synthesis. Oxylipins stimulate thiol synthesis and the subsequent increase of the antioxidant pool or regulate protein activity of detoxifying enzymes to ameliorate oxidative stress. Legend: green arrows: positive impact; red arrows: negative impact.

**Table 1 antioxidants-12-00814-t001:** Interaction of oxylipins with proteins of ROS synthesis/scavenging and the redox network. Oxylipins modulate protein amounts of the redox network as well as their catalytic activity. Affected proteins include essential proteins of different functional categories of the redox network, such as TRXs as redox transmitter, as well as ROS scavenging enzymes such as peroxidases. Stimulatory effects of oxylipins on protein synthesis/activity is denoted as “↑”, inhibitory effects as “↓” and not yet characterized effects of detected oxylipin binding as “?".

ROS and Redox	Protein	TAIR	Regulation by Oxylipins		Reference
Network			Synthesis	Activity	
Redox transmitter	TRX	At5g42980 At1g45145 At3g02730 At3g15360		↓OPDA	[56,133]
	GRX	At1g28480 At5g40370 At4g28730	↑OPDA	↓OPDA	[56,66]
Redox sensor	PRXIIB	At1g65980		↓OPDA	[56]
	2-CysPRX	At3g11630		↓OPDA	[133]
	APX	At1g07890		↑4-HNE, acrolein	[118,138]
	GPX	At4g11600	↑OPDA		[66]
	GAPDH	At1g12900	↓OPDA ↑OPDA	↓OPDA ? 4-HNE	[56,103,129]
		At1g79530			
		At3g04120 At1g13440			
Redox target protein	Cyp20-3	At3g62030		↑OPDA ? 4-HNE	[28,118,133]
	Cysteine synthase	At3g59760	↑OPDA	? 4-HNE	[66,118]
	GSTs	At2g29450 At1g02930 At2g30860	↑OPDA, acrolein, MDA	? 4-HNE	[66,71,118,119,129]
	HSP	At3g12580 At3g46230 At5g12020 At4g10250 At5g12030 At1g525690	↑OPDA, PPA1	? 4-HNE	[66,103,118,119,136]
	FBPase	At1g43670		↓acrolein	[102]
	DHAR2	At1g19570	↑OPDA		[66,129]
	MDHAR	At1g63940		? 4-HNE	[103]
ROS synthesis	Mn SOD	At3g10920		? 4-HNE	[103,118]
	Cu/Zn SOD	At2g28190 At1g08830 At2g28190 At5g23310	↓OPDA	↑ 4-HNE, acrolein ↓4-HNE, acrolein	[66,136,138]
	RBOHs	At5g47910 At1g64060	↑4-HNE, acrolein ↓4-HNE, acrolein	↑ OPDA, 4-HNE, acrolein	[96,138]
ROS scavenging	Catalase	At4g35090 At1g20630	↑OPDA	↑4-HNE, acrolein	[118,129,138]
	Peroxidase 34	At3g49120		? 4-HNE	[118]

## Data Availability

Data sharing not applicable.

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
