# Peer review of "Oxylipins and Reactive Carbonyls as Regulators of the Plant Redox and Reactive Oxygen Species Network under Stress"

_antioxidants, 2023, doi:10.3390/antiox12040814_

Round 1

Reviewer 1 Report

The paper titled "Oxylipins and reactive carbonyls as regulators of the plant redox and reactive oxygen species network understress" reviewed the ROS generation, control and subsquent stress defense signaling pathways in plants.

1. In plant pathogen part: from line 555, the accumulation of ROS is a sign of plant defense response, normally in the pattern-triggered immunity, the hypersensitive response is one part, if possible, the author can discuss more about this.

2.Is there any redox-regulatory pathway happen when pathogen invasion?

Author Response

  1. In plant pathogen part: from line 555, the accumulation of ROS is a sign of plant defense response, normally in the pattern-triggered immunity, the hypersensitive response is one part, if possible, the author can discuss more about this.

This is a very interesting suggestion. We have added an additional part to chapter 4.3.2 “Pathogen infection and induced systemic resistance (ISR)” about known and hypothetical links of oxylipins to ROS burst and hypersensitive response (see lines 619-652). This text also includes the positive effect of 4-HNE and acrolein on HR-driven PCD formerly described in ll. 444-446.

  1. Is there any redox-regulatory pathway happen when pathogen invasion?

To our knowledge, redox-regulatory pathways triggered by pathogen invasion, which might be diverse dependent on pathogen and plant species, have not been characterized in detail. As involvement of TRX is strongly implied (Mata-Pérez 2019), we have included this question in the additional text mentioned above. However, unfortunately, no explicit redox-regulatory pathway could be described.

Reviewer 2 Report

This review is summarized of the current knowledge on the interaction of the redox regulatory network and oxylipins. Also, this manuscript (Ms) described recent findings on the contribution of oxylipins to environmental acclimatization of plants.

I found several similar recent reviews on this topic:

1. Savchenko, Tatyana & Zastrijnaja, O & Klimov, Vyacheslav. (2014). Oxylipins and plant abiotic stress resistance. Biochemistry (Moscow). 79. 362-75. 10.1134/S0006297914040051.

2. Mano, J.; Biswas, M.S.; Sugimoto, K. Reactive Carbonyl Species: A Missing Link in ROS Signaling. Plants 20198, 391. doi.org/10.3390/plants8100391

3. Dreyer A.; Dietz K.-J. Reactive Oxygen Species and the Redox-Regulatory Network in Cold Stress Acclimation. Antioxidants 20187, 169. doi.org/10.3390/antiox7110169

4. Sharma P., Jha A.B., Dubey R.S., and Pessarakli M. Reactive Oxygen Species, Oxidative Damage, and Antioxidative Defense Mechanism in Plants under Stressful Conditions. Journal of Botany 2012, Article ID 217037, doi:10.1155/2012/217037

It should be noted that the authors have created a unique review with the latest data about lipid peroxidation, subsequent formation of oxylipins and their highly diverse functions in basic plant processes.

Decision: - Accept after minor revisions (which the authors can be trusted to make)

- Minor Revisions

1) line 22: reactive oxygen species) => reactive oxygen species

2) lines 172:3.1(. Non-) enzymatic lipid peroxidation… => Non-enzymatic lipid peroxidation…

3) line 176: substrate: In plants, … => substrate: in plants

4)  line 177: eicosapentaenoic acid => eicosapentaenoic acid

5) line 209: up to 50fold => up to 50-fold

6) line 259: in A. thaliana => in Arabidopsis thaliana

7) line 269: of retrograde signaling. . => of retrograde signaling.

8) line 279: Arabidopsis, Camelina microcarpa, => A. thaliana, Camelina microcarpa,

9) line 600: in Z. mays => in Zea mays

10) make a list of references according to the requirements of the journal.

Author Response

Minor Revisions

1) line 22: reactive oxygen species) => reactive oxygen species

Suggested change was made.

2) lines 172:3.1(. Non-) enzymatic lipid peroxidation… => Non-enzymatic lipid peroxidation…

Suggested change was made, caption was changed to “Non-enzymatic and enzymatic”.

3) line 176: substrate: In plants, … => substrate: in plants

Suggested change was made.

4)  line 177: eicosapentaenoic acid => eicosapentaenoic acid

Suggested change was made.

5) line 209: up to 50fold => up to 50-fold

Suggested change was made.

6) line 259: in A. thaliana => in Arabidopsis thaliana

Suggested change was made.

7) line 269: of retrograde signaling. . => of retrograde signaling.

Suggested change was made.

8) line 279: Arabidopsis, Camelina microcarpa, => A. thaliana, Camelina microcarpa,

Suggested change was made.

9) line 600: in Z. mays => in Zea mays

Suggested change was made.

10) make a list of references according to the requirements of the journal.

Suggested change was made: List of references is created using the citation style: “Antioxidants MDPI” (program: Citavi 6), adjusted position of the heading (left-bound) and style (“MDPI_2.1_heading1”) and style of cited literature (“MDPI_7.1_references”)